# Coagulation pretreatment could deteriorate reverse osmosis membrane fouling

Haojie Ding [1], Shuai Liang [2] ✉, Weichen Lin[1], Chao Chen[1], Ruonan Gao[2], Yufang Li[1], Ye Li[3], Kang Xiao [4] ✉ & Xia Huang [1] ✉

Coagulation is widely regarded as an indispensable pretreatment process in reverse osmosis (RO) systems of zero liquid discharge applications. Yet in practical applications, coagulation pretreatment often causes perplexing impact on membrane fouling and even deteriorates the RO performance with ambiguous mechanisms, thereby seriously disrupting the progress of RO-based applications. This study systematically reveals the RO performance devolution caused by Fe- or Al-based coagulation pretreatment, and elucidates the fundamental mechanism of membrane fouling deterioration due to residual coagulants. The Al-based coagulation predominantly triggers inorganic fouling, with the disruption of microbial ecological interaction networks within the biofilm exacerbated by copper-induced oxidative stresses. The Fe residues dramatically enhance the production of extracellular polymeric substances and facilitate robust fouling layer development, exacerbating membrane fouling and diminishing RO performance. These findings not only provide essential engineering guidance for optimizing practical operations but also deepen the understanding of the coagulation–RO interactions, establishing a refined framework for enhancing the efficiency and sustainability of advanced water treatment systems.

Zero liquid discharge (ZLD) is a critical strategy for industrial wastewater treatment, driven by the growing need for sustainable water reuse under global water scarcity[1,2]. Industrial wastewater poses significant environmental hazards due to its large discharge volume, high salinity, and abiotic affinity. To date, increasingly stringent regulations have compelled ZLD to become the mainstream strategy for industrial brine management[3,4]. The multi-membrane integrated process based on reverse osmosis (RO) is one of the most promising ZLD technologies, and many developed countries and emerging economies, particularly China, are vigorously promoting its application. However, the baffling fouling of RO membranes constitutes a crucial impediment to the stable operation of the ZLD system, urgently demanding fundamental research to alter this situation.

Previous endeavors on RO membrane fouling issues mainly stress emphasis on the design of anti-fouling materials[5–8], optimization of operation processes[9–12], and development of cleaning agents[13–15]. Nevertheless, from the perspective of the whole ZLD process, the scientific understandings of the overall process control and internal interaction mechanism are still vague. In the multi-membrane integrated processes, coagulation pretreatment is widely recognized as an effective approach to alleviate RO fouling. A general cognition is that the coagulation can remove suspended solids, colloidal particles[16], dissolved natural organic matters[17–19], and transparent exopolymer particles[20], thereby reducing the burden of subsequent membrane fouling. This recognition mainly derives from seawater desalination, where coagulation combined with sand filtration has been widely and successfully applied.

[1]State Key Laboratory of Regional Environment and Sustainability, School of Environment, Tsinghua University, Beijing, China. [2]Beijing Key Lab for Source Control Technology of Water Pollution, College of Environmental Science and Engineering, Beijing Forestry University, Beijing, China. [3]China Huaneng Group Clean Energy Research Institute Co., Ltd, Beijing, China. [4]College of Resources and Environment, University of Chinese Academy of Sciences, Beijing, China. ✉e-mail: shuai_liang@bjfu.edu.cn; kxiao@ucas.ac.cn; xhuang@tsinghua.edu.cn

In industrial ZLD systems, pretreatment often relies on low-pressure membranes rather than sand filtration (Supplementary Fig. 1). This makes coagulant residuals more likely to reach the RO system, and practical engineering operations have already shown that such pretreatment can cause a perplexing impact on membrane fouling and even deteriorate RO system performance[21–24]. These unexpected effects are not fully understood, with limited insight into how coagulants contribute to the fouling process under real ZLD conditions. Although residual coagulants are suspected to induce complex fouling behaviors through coupled microbial and physicochemical interactions, the underlying mechanism remains poorly understood, and there are scarce reports in the literature. Addressing this gap is essential for resolving persistent operational challenges and rethinking upstream pretreatment strategies.

Herein, we established an RO-based process for the treatment of practical desulfurization wastewater, and systematically revealed the fundamental mechanism of why coagulation pretreatment could deteriorate RO membrane fouling based on two most typical inorganic coagulates: Fe- and Al-based salts. We elucidated the dynamic development of inorganic fouling, organic fouling, and biofouling on RO membranes under different pre-coagulation scenarios, and disclosed the impact of residual coagulant on microbial community structures and metabolic functions in the fouling layers. Our findings potentially subvert the conventional cognition of coagulation pretreatment in RO research and application, and fill the current knowledge gap about membrane fouling control theory. We envision that this study will support the revision of the basic design and operation principles of RO-based ZLD applications.

## Results

### Residual coagulants deteriorate RO performance

Although coagulation pretreatment can effectively reduce the load of organic matter, it will unavoidably bring about residual metals to the feed of the RO system, which has an uncertain effect on RO membrane fouling. Our findings indicated that, due to the residual metals in the feed water, both the most typical Al-based and Fe-based coagulation pretreatments would lead to the altered fouling layer structures with different microbiota networks (Fig. 1a), resulting in deteriorated RO performance.

Three coagulation pretreatment scenarios, including the Ctrl (i.e., without coagulation), Al (i.e., coagulation with $AlCl_3$), and Fe (i.e., coagulation with $FeCl_3$) scenarios, were systematically compared. As shown in Fig. 1b, the initial flux for each scenario was controlled at ~25 L m$^{-2}$ h$^{-1}$. A sharp flux decline was observed in all three scenarios during the first six days before a pseudo-stable state was reached. At the end of the 20-d operations, decreases in flux of $25.5 \pm 0.6\%$, $49.3 \pm 0.2\%$, and $71.2 \pm 0.5\%$ were observed in the Ctrl, Al, and Fe scenarios, respectively. These results directly indicated that both the Al-based and Fe-based coagulations exacerbated the RO membrane fouling during the long-term operations, and the membrane in the Fe scenario suffered the worst fouling.

To clarify whether this disparity could be attributed to ion accumulation caused by concentration polarization (CP), theoretical estimations showed minimal differences in enrichment factors among the Fe, Al, and Ctrl groups, with values of 1.15, 1.18, and 1.16, respectively (Supplementary Note 1). These similarly low values suggest that CP alone was unlikely to drive the overall fouling differences, although it may still contribute locally in the presence of residual coagulant-derived substances.

### Coagulation pretreatment induces thicker fouling layer with discrepant structures

Thickness measurements by scanning electron microscopy (SEM) quantitatively revealed the disparities in fouling layer thickness among the three scenarios (Fig. 1d). After the 20-d operations, the mean thicknesses of the fouling layers in the Al and Fe scenarios were separately measured to be $36.2 \pm 11.6$ μm and $76.2 \pm 19.7$ μm, significantly exceeding the $13.4 \pm 7.1$ μm observed in the Ctrl scenario ($p < 0.05$). Such thickness increases directly correlated with the observed reductions in permeate flux, further underscoring the critical role of fouling layer structure in influencing membrane performance, as consistent with findings in previous studies[25,26]. In addition, optical inspection showed that the fouling layers in the Al and Fe scenarios exhibited yellow-brown and reddish-brown colors, respectively, and were accompanied by dense aggregation (Fig. 1c). Detachment and regrowth were observed after 8 days, corresponding to the fluctuations in filtration resistance (Supplementary Fig. 2).

The top-view SEM images also revealed pronounced differences in fouling morphology. At the 12 h during the operation, the Ctrl membrane retained most of the original surface with minimal inorganic particle attachment (Fig. 1e), whereas the membranes in coagulants-pretreated groups exhibited tightly adhered inorganic particles and bacteria, leading to the early rapid flux declines (Fig. 1b), as supported by the energy dispersive spectrometry (EDS) elemental distribution shown in Supplementary Fig. 3. In the subsequent operations (6–20 d), the fouling layer in the Ctrl group remained relatively porous, which contributed to reduced hydraulic resistance and a relatively higher flux. In contrast, the membrane surface of the Al group showed bacterial particles dispersed within inorganic scaling, with fewer surrounding organic matter; and the surface of the Fe group exhibited notable accumulation of bio-, organic-, and inorganic foulants, which suggested that filamentous and layered extracellular polymeric substances (EPSs) facilitated bacterial aggregation on inorganic particles[27–29]. This contrast in fouling morphology reflects distinct patterns of interaction between microbes and deposited substances in the two groups.

Furthermore, it could be hypothesized that during the operation, CP might have led to the progressive binding of inorganic scaling to EPSs, which likely displaced bacterial colonies, resulting in cell damage and perforation[30–33]. As observations in Fig. 1e suggest, the damaged bacterial structures could function as new nucleation sites for inorganic substances, while the lysed cell debris could be utilized as nutrients by newly generated cells[34]. Consequently, a densely cross-linked three-dimensional fouling layer was developed on the Fe-group membrane surface, as compactness being a crucial factor in flux decline[35].

### Residual Fe induced more organic and biological fouling

The observed distinction in fouling layer structures can be primarily attributed to the variations in chemical compositions of the foulants deposited on the membrane surface. The variations of inorganic content are depicted in Fig. 2a–c. High deposition levels were observed for iron, copper, silicon, aluminum, and zinc, aligning with the EDS elemental distribution results (Supplementary Fig. 3). Copper was the highest deposited element in both the Ctrl and Al scenarios, and the slight difference in its deposition between the Al and Fe groups suggests the influence of fouling composition and competition between metal ions for binding sites beyond feed concentration (Supplementary Table 1). In the Fe scenario, the iron deposition peaked at 295 μg cm$^{-2}$ on 6 d, which was 8.5-fold and 17.8-fold higher than those in the Al and Ctrl scenarios, respectively. Elevated iron levels have been shown to promote biofilm development by acting as an electron acceptor and facilitating electron transfer under oxygen-limited conditions[23]. Besides, soluble $Fe^{2+}$ could act as a cofactor for the ferric uptake regulator[36–38] (Fur, a key protein that controls the expression of genes involved in iron uptake and metabolism), thereby significantly influencing the iron homeostasis within the biofilm[39]. This could be supported by the X-ray photoelectron spectroscopy (XPS) analyzes, which revealed a $Fe^{2+}/Fe^{3+}$ ratio of 1.27 in the Fe group, higher than the 0.64 in the Ctrl group (Supplementary Fig. 4). The pattern of iron

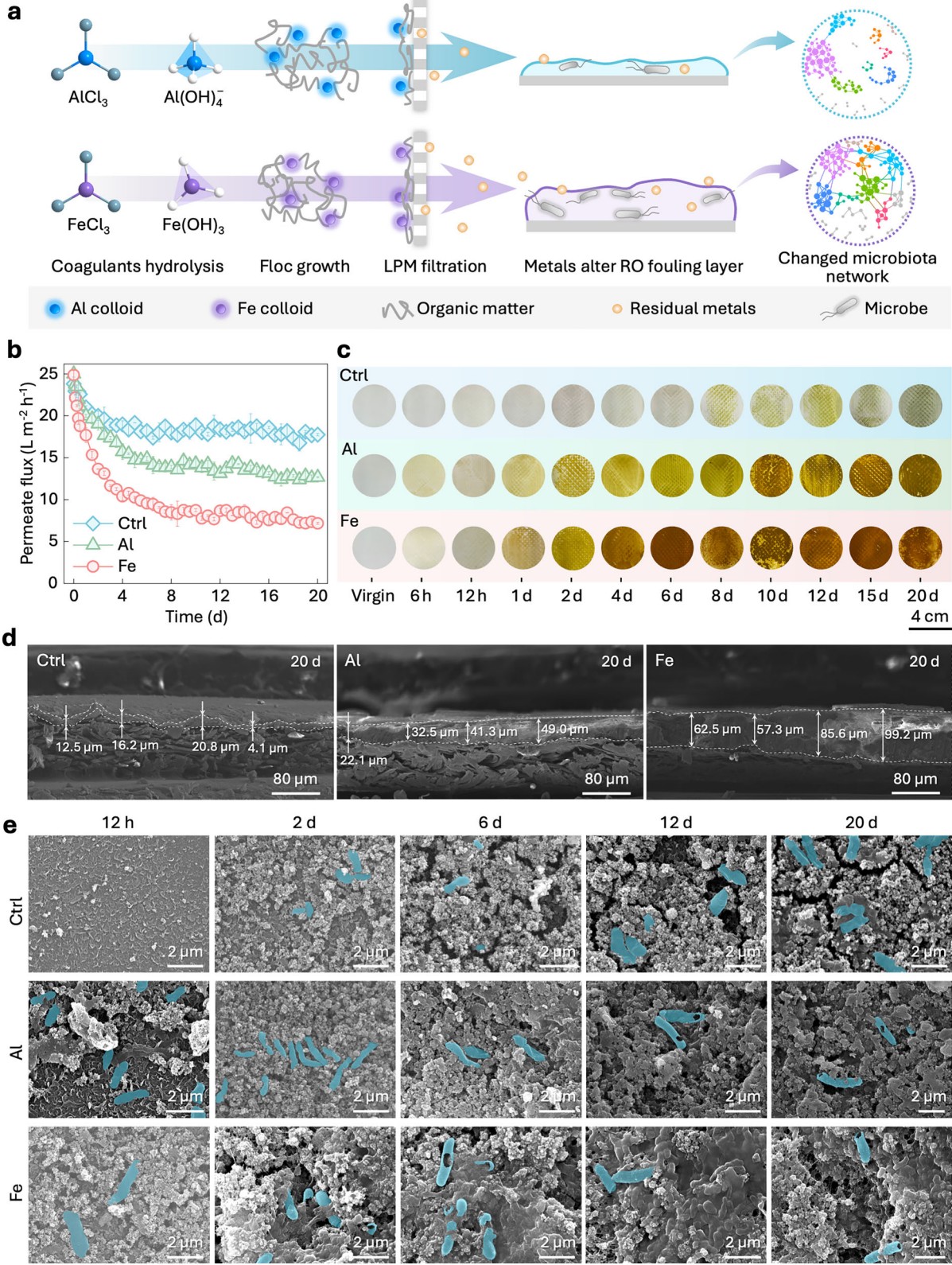

**Fig. 1 | Coagulation-caused deterioration of reverse osmosis (RO) performance.** **a** Schematic illustration of the deteriorated RO membrane fouling caused by residual coagulants after the low-pressure membrane (LPM) units. **b** Comparison of the flux variations during a 20-d operation among three different coagulation pretreatment scenarios: Ctrl (i.e., control, without coagulation), Al (i.e., Al-based coagulation), and Fe (i.e., Fe-based coagulation). **c** Comparison of the membrane surface photographs at different time points during the 20-d operation. **d**, **e** Scanning electron microscopy (SEM) views of the cross-sections (**d**) and top

surfaces (**e**) of the RO membranes in the Ctrl, Al, and Fe scenarios. The residual concentrations of aluminum and iron were 2.36 mg L$^{-1}$ and 2.59 mg L$^{-1}$ in the Al and Fe groups, respectively, and the feedwater pH values were 6.84, 6.98, and 7.10 in the Al, Fe, and Ctrl groups. Cyan markers indicate regions with clearly exposed microbial shells, while areas dominated by EPS without identifiable features were not labeled. Error bars in the figures represent the s.d. ($n$ = 3), and data are presented as mean values ± s.d.

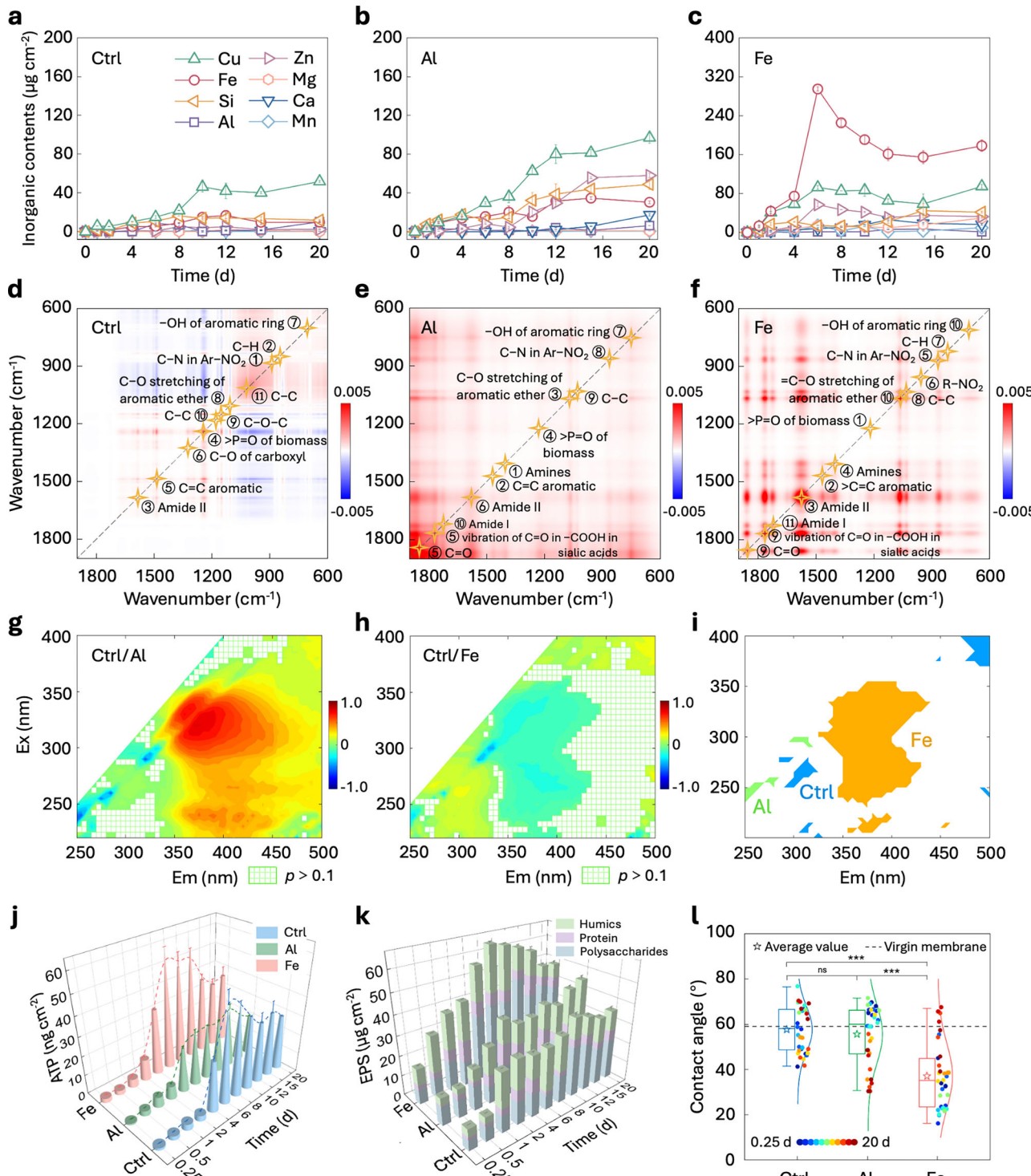

**Fig. 2 | Dynamic chemistry variation of RO fouling layers. a–c** Content variations of different inorganic elements on the RO membrane surfaces in the (**a**) Ctrl, (**b**) Al, and (**c**) Fe scenarios. **d–f** Synchronous maps of two-dimensional correlation spectroscopy (2DCoS) for organic functional groups detected in the (**d**) Ctrl, (**e**) Al, and (**f**) Fe scenarios. The serial number indicates the sequence in which functional groups emerge, and the repeated numbers correspond to the simultaneous appearance of two functional groups. **g, h** Comparison of the calculated fluorescence quotient (FQ) spectra of Ctrl/Al (**g**) and Ctrl/Fe (**h**) over the 20-d period. **i** Distribution of prominent FQ regions in the Ctrl, Al, and Fe scenarios. The

wavelength regions where $FQ_{Ctrl/Al} < 0$ and $FQ_{Al/Fe} > 0$, $FQ_{Ctrl/Fe} < 0$ and $FQ_{Al/Fe} < 0$, and $FQ_{Ctrl/Al} > 0$ and $FQ_{Ctrl/Fe} > 0$ are regarded as being Al prominent, Fe prominent, and Ctrl prominent, respectively. **j** Variations of ATP content in the fouling layers. **k** Quantitative analyses of dynamic changes in the polysaccharides, protein, and humic components. **l**, Measured contact angles of the virgin and fouled membranes (Box plot: center line indicates median; box edges indicate the 25th and 75th percentiles; whiskers extend to 1.5 × the interquartile range; $n = 11$ per group). Error bars in the figures represent the s.d. ($n = 3$) and data are presented as mean values ± s.d.

utilization can be reinforced by the microbes within the thicker biofilms.

Fourier transform infrared spectrometer (FTIR) provided detailed insights into the organic functional groups present in the fouling layers. Peaks corresponding to >P=O, –OH, and Amide I appeared and disappeared over time, indicating a complex interplay between the functional groups and temporal changes (Supplementary Fig. 5). Two-dimensional correlation spectroscopy (2DCoS) spectra further revealed subtle time-dependent variations in the fouling layer's response to coagulant residues (Fig. 2d–f and Supplementary Fig. 6). Thirteen characteristic peaks were identified in the Fe group along the diagonal, compared to the eleven peaks in the Ctrl and Al groups. In addition, peak signals in the Ctrl group were weaker, implying that the residual coagulants increased the accumulation of organic components in the fouling layers. Asynchronous spectra, according to the Noda's rule, revealed the sequence of appearance of functional groups during membrane fouling (Supplementary Note 2 and Tables 2–4)[40,41]. The Fe group showed early appearance of phospholipid >P=O of biomass (1240 cm$^{-1}$) and protein Amide II (1585 cm$^{-1}$), confirming vigorous microbial activity induced by residual iron.

Three-dimensional excitation-emission matrix (3D-EEM) analyzes showed that no noticeable discrepancy was observed in the RO feed water among the three scenarios, with consistent peaks for aromatic protein I and II in the Em = 280–350 nm range (Supplementary Fig. 7). However, the fluorescence quotient (FQ) spectra revealed significant differences in the fluorescence regions of organic matter on the RO fouling layers (Fig. 2g, h and Supplementary Fig. 8). Positive $FQ_{Ctrl/Al}$ results predominantly appeared in the Ex < 235 nm and Ex = 300–350 nm regions, suggesting that soluble microbial products and hydrophobic acids were statistically lower in the Al scenario[42]. In contrast, the Fe group exhibited significantly higher fluorescence intensity in the middle Em region, reflecting enhanced levels of plankton-derived dissolved organic matter and marine humic-like substances. Non-parametric analyses of the 3D-EEM spectra highlighted key regions of fluorescence signals among the three groups (Fig. 2i). Overall, the Ctrl group exhibited pronounced fluorescence peaks primarily in the Em = 300–330 nm range, whereas the Fe group demonstrated the most substantial features in the Em = 330–450 nm region. Conversely, the Al group was predominantly characterized by the lower Stokes shift band[43]. These findings revealed an enhanced microbial activity in the Fe scenario and a significantly suppressed activity in the Al scenario.

The ATP content can quantitatively represent the total biomass in the fouling layer. The Fe group exhibited an ATP content of 61.2 ng cm$^{-2}$ at 6 d, which was 1.27 times higher than that of the Ctrl group, while the ATP of the Al group content was only 0.55 times lower than that of the Ctrl group (Fig. 2j).

In the Fe scenario, the humic acid component within EPSs was reduced from 45.6% to 14.6%, while polysaccharides exhibited a dynamic increase from 38.6% to 73.9% over the 20-d period (Fig. 2k). This trend reflects intensified EPSs secretion driven by iron stimulation. Conversely, in the Al group, the polysaccharide content in EPSs decreased from 67.7% to 48.4%, likely due to the elevated copper deposition, as copper has been observed to be negatively correlated with bacterial EPSs content[44]. Confocal laser scanning microscopy (CLSM) images further supported these findings by visually capturing the evolution of organic and microbial components throughout the operational period (Supplementary Fig. 9).

In addition, contact angle variations revealed temporal changes in the ratio of organic to inorganic components in the fouling layers (Fig. 2l). In the Al scenario, the contact angle decreased over time, paralleled by the Ctrl scenario, indicating a gradual increase of hydrophilicity. In marked contrast, the samples in the Fe scenario initially showed a higher hydrophilicity, which then shifted towards hydrophobicity as the organic content increased. This highlights the substantial impact of organic accumulation in altering the fouling layer's surface properties, particularly its hydrophilicity and hydrophobicity[45,46].

## Shifts in microbial community and key functional genes

Variance partitioning analysis based on multiple linear regression was performed to assess the individual and interactive contributions of the biological, inorganic, and organic fouling to the flux decline (Fig. 3a). Individual fouling contributions varied, with organic fouling being most prominent in the Ctrl group (18.3%) and biofouling leading in the Al group (15.1%), while single-factor effects were less significant in the Fe group. At the binary interaction level, biological-inorganic factors dominated in the Al (28.9%) and Ctrl (42.0%) scenarios, whereas inorganic-organic interactions prevailed in the Fe scenario (49.1%). However, in triadic interactions, the Fe scenario showed the highest contribution (39.0%), indicating that microbes play a more substantial role as the complexity of fouling increases. These findings underscore the pivotal role of microbes in shaping the distinct progression of complex fouling types when higher-order interactions are involved. Based on full-scale RO operational data, the on-site validation presented in Supplementary Fig. 10 shows that the triadic interactions dominate and support the applicability of the partitioning analysis under real ZLD conditions, confirming the consistency with laboratory results.

To elucidate the interactions between residual coagulants and RO biofouling, high-throughput sequencing was deployed to dissect microbial community structures at various temporal stages for each experimental group. The operational taxonomic units (OTUs) were 1354, 621, 783, and 1169 for the Feed, Ctrl, Al, and Fe groups, respectively (Fig. 3b). Compared to the feed water, the microbial diversity and richness in the RO fouling layers were markedly diminished (Fig. 3b and Supplementary Fig. 11), inferring selective colonization of microbial species on the RO membranes. Among the three RO fouling layers, the Fe group harbored the highest number of 364 unique OTUs, substantially exceeding the 90 and 111 OTUs observed in the Al and Ctrl groups, respectively. This hinted the emergence of a distinct ecological niche in the Fe group. Principal coordinates analysis further uncovered pronounced clustering and dispersion patterns among the different scenarios (Fig. 3c), indicating that the presence of coagulant residues altered the trajectory of community structure succession.

Taxonomic analysis of community structure illuminated specific bacterial dynamics. At the family level, *Pseudomonadaceae* and *Burkholderiaceae* were predominant in the Feed group (Supplementary Fig. 12). In the Fe and Al fouling layers, *Burkholderiaceae* exhibited superior adaptability compared to *Pseudomonadaceae*, whereas the opposite trend was observed in the Ctrl group. This implied selective colonization of microbial taxa on the RO membranes mediated by residual coagulants. Furthermore, *Xanthobacteraceae*, *Cellulomonadaceae*, and *env. OPS_17* were insignificantly dominated in the feed water, demonstrated increased abundance in the Fe scenario, highlighting the unique habitat structure of the Fe biofilm.

As shown in Fig. 3d, the genus-level microbial community underwent profound transformations across the three-types of fouling layers. *Pseudomonas*, which comprised 77.5 ± 6.5% of the Feed group, surged to 90.3 ± 0.9% on the RO membrane in the Ctrl group. Conversely, its relative abundance in the Fe and Al groups plummeted from 72.7 ± 3.9% and 44.6 ± 1.6% to below 0.3%, respectively. This emphasizes *Pseudomonas*'s dominant role in RO membrane fouling and its principal contribution to EPSs secretion in the Ctrl group[47]. In marked contrast, *Sphingopyxis*, known to be strongly linked to biofouling[15,48,49], demonstrated exceptional adaptability, with its relative abundance rising from < 0.1% in the influent to 34.9 ± 0.9%, 29.7 ± 1.1%, and 47.5 ± 17.8% in the Ctrl, Fe, and Al groups, respectively. Moreover, the genus *Limnobacter* was primarily detected in the Ctrl

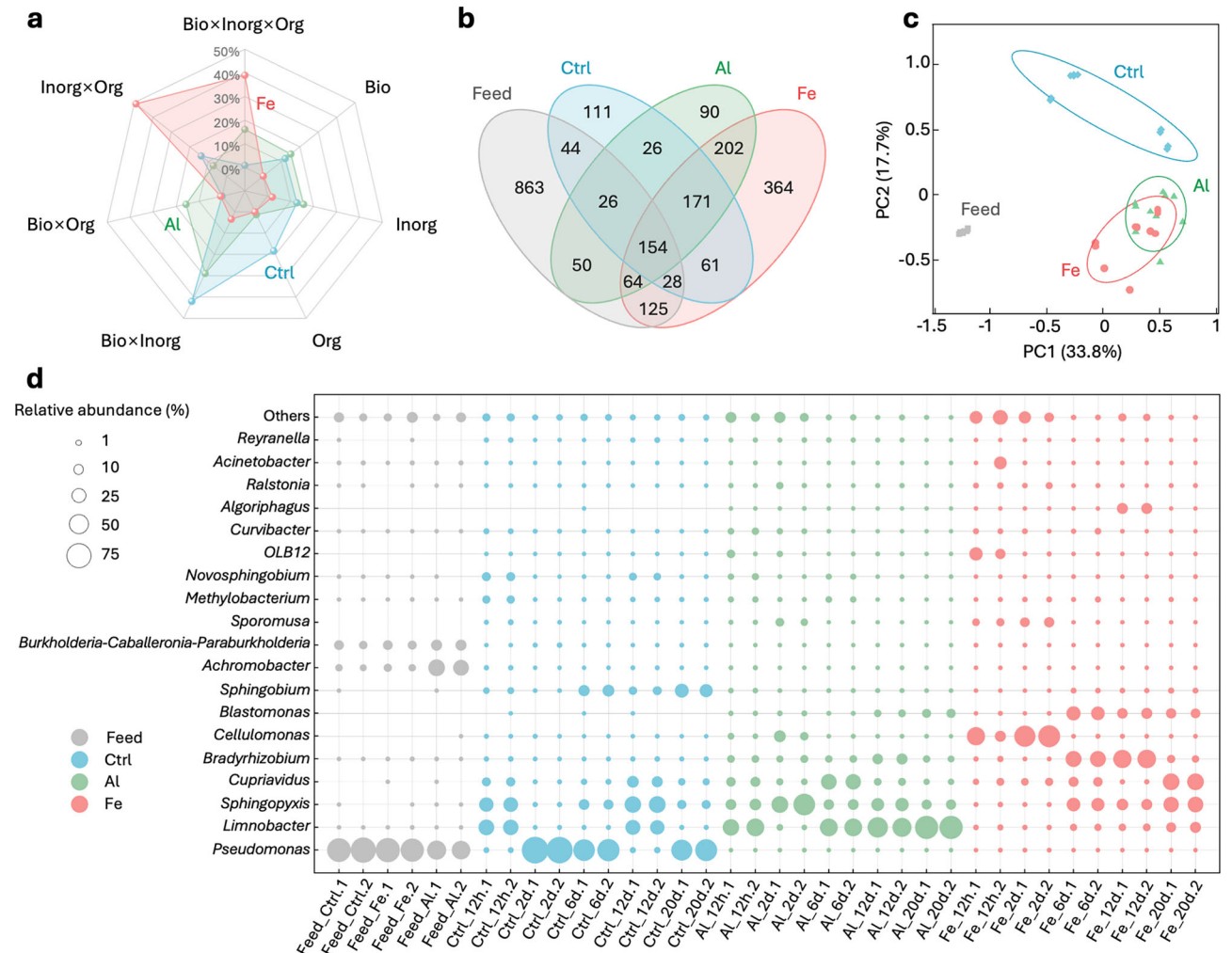

**Fig. 3 | Biofouling contributions and microbial community characterizations.**
**a** Individual and interaction contributions of the fouling factors based on variance partitioning analysis. **b** Venn diagram on the unique and shared microbial operational taxonomic units (OTUs) between feed and RO membrane samples. **c** Principal coordinates analysis of feed and biofilm samples. **d** Relative abundances of dominant microbial at the genus level.

and Al groups, pointing to coagulant conditions selectively favor its proliferation.

Persistent genera that produce EPSs[35,50,51], including *Bradyrhizobium, Cupriavidus, Blastomonas, Sphingopyxis,* and *Cellulomonas,* were markedly enriched in the Fe group, where environmental factors substantially promoted microbial growth. *Bradyrhizobium* stands out for its ability to efficiently sequester iron via iron-binding proteins and to upregulate genes associated with iron uptake, thereby intensifying membrane fouling within the Fe group[52]. Similarly, *Cupriavidus,* renowned for its copper resistance, augments copper adsorption through cell surface proteins, thereby mitigating copper influx into the cells[51]. These increased presence of *Sphingopyxis* and *Bradyrhizobium* significantly stimulated polysaccharide secretion and biofilm formation. In contrast, the correlation heatmap illustrated a negative correlation between heavy metal concentrations and most microbial abundance in the Al group (Supplementary Fig. 13). These findings underscore the divergent impacts of metal residues on microbial community composition and fouling behavior.

Redundancy analysis was performed to further dissect the influence of physicochemical parameters. As shown in Supplementary Fig. 14, the redundancy analysis (RDA) axes RDA1 and RDA2 explained 98.9%, 68.0%, and 90.3% of the total microbial variation in the Ctrl, Fe, and Al groups, respectively. Iron and copper were identified as the most significant inorganic metal factors influencing community

structure, with *Pseudomonas, Sphingopyxis, Limnobacter, Bradyrhizobium,* and *Cupriavidus* being the most affected species.

To further elucidate the functional dimensions of microbial communities, phylogenetic analysis leveraging random matrix theory was performed to uncover the ecological interaction networks within these communities (Fig. 4a–c and Supplementary Table 5). Overall, the Fe group demonstrated enhanced stability, characterized by a reduced network size and the highest number of modules exhibiting exceptional modularity and system efficiency[53]. In contrast, the Al group displayed fewer modules and total nodes compared to the Ctrl group, reflecting network perturbations under heavy metal stress. Specifically, g_*Sphingopyxis*, o_*Rhizobiales*, and p_*Proteobacteria* functioned as network connectors in the Fe group (Zi < 2.5, Pi > 0.62), whereas o_*Planctomycetales* served as module hubs (Zi > 2.5, Pi < 0.62) (Fig. 4f). In comparison, only f_*Stenotrophomonas* acted as a module hub in the Ctrl group, with no connectors or hubs identified in the Al group (Fig. 4d, e). These findings emphasized the superior stability of the microbial ecological interaction network in the Fe group, which correlates with more pronounced biofouling.

To investigate the key functional characteristics of microbial communities within the RO fouling layers, metabolic pathways were analyzed using the Kyoto Encyclopedia of Genes and Genomes (KEGG) database (Supplementary Table 6). Metabolism emerged as the dominant function, comprising approximately 48% of the KEGG

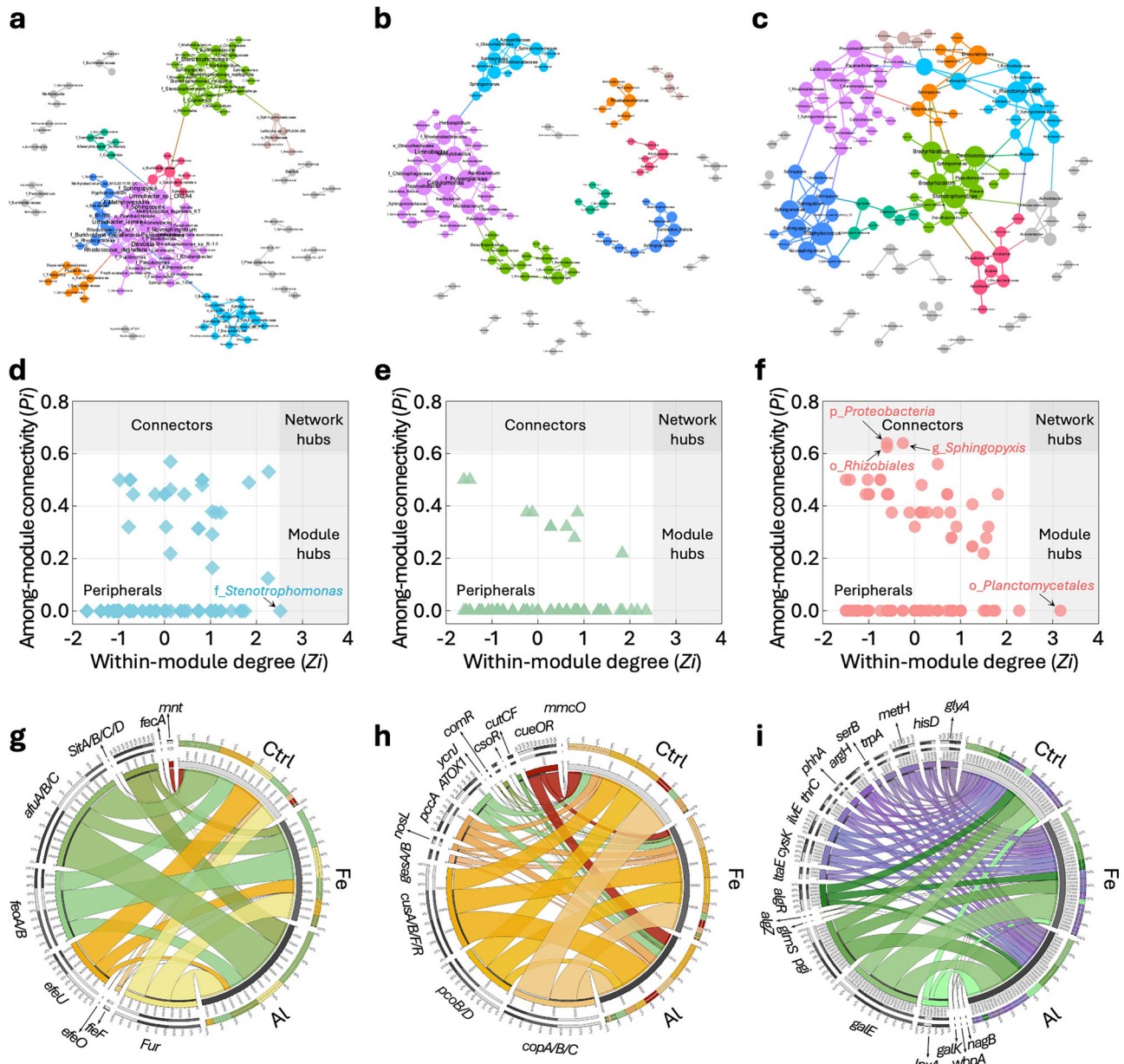

**Fig. 4 | Microbial network and key functional genes. a–c** Network interactions in the RO fouling layers of the Ctrl (**a**), Al (**b**), and Fe (**c**) groups. **d–f** The species topological roles of the Ctrl (**d**), Al (**e**), and Fe (**f**) groups based on the Pi-Zi plots. **g–i** Iron metabolism-related (**g**), copper metabolism-related (**h**), and exopolysaccharide and amino acid biosynthesis-related (**i**) functional genes in the Ctrl, Al, and Fe groups. In (**a–c**), the size of nodes reflects the corresponding degree, and different colors indicate different modules. In (**g**), yellow, green, and red bands indicate Fe acquisition, transport, and reduction genes, respectively. In (**h**), yellow, green, and red bands represent copper resistance/efflux, transport/repression, and oxidation genes, respectively. In (**i**) green bands denote polysaccharide-related genes, and purple bands indicate amino acid- related genes.

pathways at level 1. At KEGG level 2, remarkable discrepancies were observed between the Fe and Al groups relative to the Ctrl group, with Fe and Al displaying 15 and 17 significantly distinct pathways, respectively ($p < 0.05$) (Supplementary Fig. 15). The Fe group displayed a marked enhancement in pathways related to amino acid metabolism, xenobiotics biodegradation, carbohydrate metabolism, metabolism of other amino acids, and global overview, compared to the Al group. By contrast, the Al group showed diminished pathways in membrane transport, signal transduction, and nucleotide metabolism relative to the Ctrl group. Notably, bacterial intracellular reactive oxygen species (ROS) levels in the Al group were 1.50-fold higher than those in the Ctrl group, accompanied by significantly elevated superoxide dismutase (SOD) and catalase (CAT) enzyme activities (Supplementary Fig. 16).

These results confer that the elevated copper concentrations in the Al group induced oxidative stress, thereby impairing bacterial metabolic processes and inhibiting growth by disrupting membrane transport and signal transduction[54,55]. In contrast, the iron stimulation in the Fe group fostered enhanced amino acid and carbohydrate metabolism, bolstering microbial activity and EPSs secretion, which alleviated the detrimental effects of copper.

To delve deeper into the intrinsic interactions between heavy metals and RO biofouling, the KEGG database was utilized to categorize genes related to iron metabolism, copper metabolism, and EPSs secretion (Fig. 4g–i). Compared to the Ctrl group, the Fe and Al groups showed upregulation of iron acquisition genes *irr* (Fur family) by 4.34 and 3.13 times, respectively (Fig. 4g). Furthermore, the Fe group

further exhibited markedly elevated levels of the iron acquisition genes *fie*F and *efe*O, with relative abundances 1.28 and 3.71 times higher than those in the Al group. In addition, the Fe group displayed 2.00-fold to 3.22-fold increases in iron transport genes, including *efe*U, *feo*A/B, *sit*A/B/C/D, and *mnt*A/C/D/R, compared to the Al group. Remarkably, the iron reduction gene *fhu*F was found to be 54.76 times more abundant in the Fe group than in the Al group, which elucidates the higher $Fe^{2+}/Fe^{3+}$ ratio observed in the Fe fouling layer.

Furthermore, the Fe and Al groups exhibited dramatic increases in copper efflux genes *ges*A/B, with relative abundances 187 and 89.7 times higher than those observed in the Ctrl group, respectively (Fig. 4h), reflecting a highly efficient copper expulsion mechanism under elevated copper stress. Despite comparable copper deposition levels across the fouling layers, the Al group displayed significantly lower relative abundances of copper resistance genes (*cop*A/B/C, *pco*B/D), copper efflux genes (*ges*A/B, *nos*L), copper inhibition genes (*com*R, *cso*R), and the copper transport gene *ycn*J, ranging from only 0.26 to 0.65 times those in the Fe group. This proves a marked deterioration in protective mechanisms in the Al group under copper stress. Further examination of EPSs secretion genes (Fig. 4i) revealed substantial upregulation in the Fe group compared to the Al group, with EPSs polysaccharide-related genes (*wbp*A, *gal*K, *gal*E, *glm*S, *alg*Z, *alg*R) elevated by factors ranging from 1.13 to 45.4, and amino acid-related genes (*lta*E, *cys*K, *phh*A, *serB-pls*C, *met*H) increased by 1.12 to 2.02 times. These findings highlight the decisive role of EPSs secretion in fortifying the Fe group against heavy metal-induced damage.

### Residual coagulants mediated metabolic mechanisms

These divergent metabolic responses are vividly depicted in Fig. 5, where the right side illustrates the vigorous metabolic activity in the Fe scenario, characterized by enhanced iron uptake, EPSs production, and efficient copper efflux, all contributing to biofilm stability. The presence of residual coagulants, particularly iron, elicits distinct metabolic adaptations within microbial communities on the RO membranes. In the Fe group (iron- and copper-rich environment), the microbes intensify iron uptake mechanisms and activate amino acid and carbohydrate metabolic pathways, culminating in the robust synthesis of EPSs and biofilm formation. This adaptive response is fortified by the upregulation of antioxidant defenses, which effectively mitigate oxidative stress and preserve cellular functionality under metal stress[56]. These findings suggest that upstream coagulation processes should avoid introducing excess biologically active metal residues that promote EPS overproduction, and emphasize the importance of integrating biological considerations into pretreatment design.

Conversely, the left side portrays the Al scenario, where microbes contend with copper toxicity and limited access to essential metals, leading to elevated oxidative stress. This cascade of stress induces DNA damage and compromised metabolic functions, progressively exacerbating microbial dysfunction and intensifying membrane fouling. These insights reveal the critical role of residual coagulants in shaping microbial metabolic pathways, implying that targeted manipulation of these residuals could fortify microbial resilience and thus exacerbate RO fouling processes in ZLD scenarios.

## Discussion

This study unveils the profound impact of coagulation pretreatment on RO performance, specifically highlighting how Fe and Al residues contribute to the evolution of the fouling layer composition, providing critical insights into optimizing fouling control strategies for ZLD operations. The Al-based coagulants primarily induced inorganic fouling and biofouling, with residual heavy metals in the feed water, particularly copper, triggering oxidative stress in microbes and inhibiting biofilm development. However, the influence of other heavy metals or pollutants should also be considered, as they may produce similar effects on oxidative stress and biofilm formation. In contrast, residual iron significantly enhanced the production of EPSs and promoted robust biofilm formation, thereby exacerbating biofouling and diminishing RO system performance. These findings highlight the critical importance of real-time coagulant control and dynamic dosing adjustments, as well as the careful selection of low-fouling coagulants in ZLD systems. Instead of merely optimizing dosage, the focus should shift towards integrating smart technologies for residual tracking and exploring targeted control strategies. In particular, given that practical ZLD operations often monitor conductivity and hardness but neglect organic indicators, tracking feedwater organic content or assimilable organic carbon could offer actionable signals to refine coagulant dosing. In addition, considering alternative coagulant types can help minimize adverse effects on membrane performance.

It should be noted that this study deliberately focused on a low-pressure microfiltration pretreatment scenario, which was selected to represent practical operating conditions commonly encountered in existing industrial ZLD systems, under which residual coagulants are more likely to be carried over to the RO stage and induce fouling challenges. While this configuration captures realistic and representative fouling behaviors, future work is warranted to further evaluate how coagulation pretreatment influences RO membrane fouling in ZLD systems employing tighter pretreatment configurations, such as ultrafiltration or conventional sand filtration. At the same time, the present findings potentially indicate that adopting sand filtration or tight ultrafiltration to minimize coagulant carryover can be a recommended strategy for improving RO stability in practical RO-based ZLD applications.

## Methods

### RO-based process for practical wastewater treatment

The practical desulfurization wastewater was obtained from a coal-fired power plant located in Yueyang City (Hunan Province, China), and was sequentially treated at room temperature ($\sim 25\,°C$) with a RO-based treatment process, including a coagulation unit, a microfiltration system, and a RO system. Four typical iron- and aluminum-based coagulants were evaluated in preliminary jar tests, and $FeCl_3$ (residual $Fe = 2.59\,mg\,L^{-1}$) and $AlCl_3$ (residual $Al = 2.36\,mg\,L^{-1}$) were subsequently selected for RO pretreatment, as described in Supplementary Note 3, Table 7, and Fig. 17. A parallel operation without coagulant addition was conducted as a control. According to the different coagulation scenarios, the corresponding membranes and foulant specimens were categorized as the Fe, Al, and Ctrl groups, respectively.

After the coagulation, the resulted supernatant of each group was filtered with a 0.45 μm microfiltration membrane (Durapore, Millipore, USA) using a solvent filtration apparatus (Jinteng, China), and the obtained effluent was used as the RO feed. This pore size selection aligns with engineering practices observed in full-scale ZLD systems, where microfiltration with pore sizes ranging from 0.3 to 0.9 μm is commonly employed. This choice ensured that the study reflects typical industrial pretreatment conditions.

The RO system contains four parallel membrane modules (CF042A-CF, Sterlitech, USA, each with an effective filtration area of $42\,cm^2$) and a high-pressure variable-frequency pump (Hydracell, Sterlitech, USA). Prior to operation, both the RO membranes and the RO system were thoroughly rinsed. Specifically, the RO membranes (BW30-4040, DOW Filmtec, USA) were first soaked in a 25% isopropyl alcohol solution for 1 h to remove residual organic matter, and then thoroughly rinsed with ultrapure water at least three times. The resulted membranes were stored in 4 °C ultrapure water in the dark. Before each fouling filtration experiment, the RO system was alternately rinsed with sodium citrate (pH 2.5) and NaOH solutions (pH 11.5) for a total of 24 h, and then flushed with ultrapure water three times.

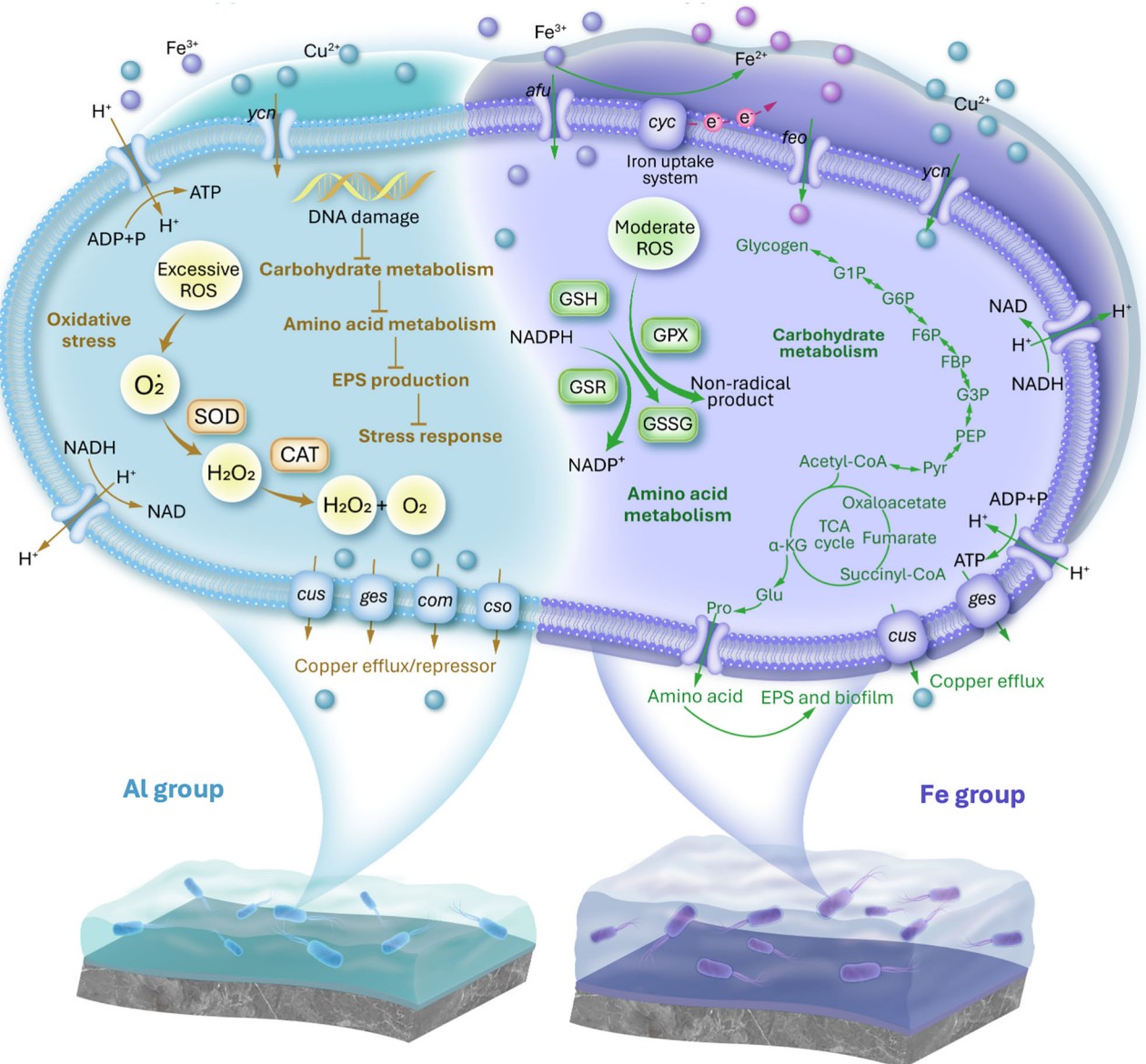

**Fig. 5 | Proposed metabolic pathways affected by residual coagulants.** Al group (left) of low-iron and copper-rich environment and Fe group (right) of iron and copper-rich environment.

## RO fouling filtration tests

The RO membranes were installed in the membrane modules with feed spacers and pre-pressurized with ultrapure water for 12 h to obtain a constant permeate flow. Subsequently, the RO fouling filtration tests were performed in a circular mode, that is, both the permeate and brine solutions were circulated back to the feed tank to ensure stable ionic strength, and the feed was replenished every 48 h. The constant applied pressure and cross-flow velocity were set at ~2.0 MPa and ~6 cm s$^{-1}$, respectively. To reveal the development of membrane fouling, the samples of the fouled membranes were sequentially collected at least in duplicate at a certain operating time (i.e., 0.25, 0.5, 1, 2, 4, 6, 8, 10, 12, 15, and 20 d), as illustrated in Supplementary Fig. 18. During the operation, the membrane flux ($J$, L m$^{-2}$ h$^{-1}$) was periodically measured under a pressure of 2.2 MPa, and calculated using Eq. (1).

$$J = \frac{\Delta V}{A \times \Delta T} \qquad (1)$$

where the $\Delta V$, $A$, and $\Delta T$ represent the permeate volume (L), effective filtration area (m$^2$), and testing duration (h), respectively.

The hydraulic resistance of the RO fouling layer ($R_f$, m$^{-1}$) was calculated by subtracting the intrinsic resistance of the virgin membrane ($R_m$, m$^{-1}$)[57] from the total filtration resistance ($R_T$, m$^{-1}$)[57], as instructed by Eqs. (2) and (3).

$$R_f = R_T - R_m \qquad (2)$$

$$R_T = \frac{\Delta P}{\mu \times J} \qquad (3)$$

where the $\Delta P$ indicates the transmembrane pressure (Pa) and the $\mu$ indicates the permeate viscosity (Pa s).

## Characterizations of membranes and fouling layers

**Membrane surface.** The morphology of membrane surfaces and element distributions of the fouling layers were determined by SEM-EDS (Merlin Compact, Zeiss, Germany). The membrane samples were pre-freeze-dried for 12 h and sputter-coated with 8 nm-thick platinum. Membrane surface hydrophilicity was evaluated in terms of water contact angle, which was measured using a contact angle instrument

(OCA20, Data Physics, Germany) on the basis of a sessile-drop method. The instantaneous profile of a 2.0 μL ultrapure water droplet set on the membrane surface was imaged and used for contact angle calculations. For each sample, the final result was averaged from at least three measurements on random locations. The distributions of different types of foulants (i.e., proteins, polysaccharides, and nucleic acids) in the fouling layer were visualized by CLSM (LSM710, Zeiss, Germany; see details in Supplementary Table 8 and Fig. 19).

**Organic components**. Membrane fouling layers were further detached from the membranes for chemistry determination. The shredded membrane sample was immersed in a 0.01 M NaOH solution and shaken for 48 h, and then the supernatant was separated with a 0.45 μm filter. The organic components were quantified using a TOC analyzer (TOC-Lcph, Shimadzu, Japan), and also characterized in terms of polysaccharides (phenol-sulfuric acid method[58,59]), protein (BCA protein assay kit, Beyotime, China), and humic substances (modified Lowry method[60]).

In addition, 2DCoS[61] coupled with FTIR spectra (Nicolet iS 50, Thermo Fisher Scientific, USA) was applied to reveal the subtle changes of the organic functional groups of the fouling layer on the synchronous and asynchronous maps. 3D-EEM fluorescence spectroscopy characterization (F-7100, Hitachi, Japan) was also performed[62]. FQ and prominent region distributions were used to clarify the statistical differences among the multiple samples in the three groups (one-tailed Wilcoxon signed rank test, $p < 0.1$, sample number $n = 11$)[63]. Data processing for the 3D-EEM and FQ characterizations was performed using MATLAB R2019a (MathWorks Inc., USA). The normalized fluorescence intensity (FI′) and $FQ_{A/B}$ in the EEM spectra were calculated using Eqs. (4) and (5):

$$FI' = 0.01 + 0.09 \frac{FI - FI_{min}}{FI_{max} - FI_{min}} \tag{4}$$

$$FQ_{A/B} = \log(FI'_A / FI'_B) \tag{5}$$

where the FI, $FI_{min}$, and $FI_{max}$ denote the original, minimum, and maximum intensities of the EEM spectra, respectively. The $FQ_{A/B}$ is defined as the logarithm of the division of spectra A by B. The distribution of the $FQ_{A/B}$ represents the difference between A and B at the given Ex/Em wavelength. The positive $FQ_{A/B}$ region represents a decrease in the relative content of organic matter from spectra A to B, and vice versa.

**Inorganic components**. To determine the inorganic components of the fouling layer, the membrane sample (~ 2.63 cm²) was immersed in a 40% nitric acid solution (5 mL) and heated for 1 h at 90 °C. The digestion solution was then filtered with a 0.45 μm filter. The contents of inorganic elements (i.e., Fe, Al, Cu, Si, Ca, Mg, Zn, and Mn) were measured by inductively coupled plasma-atomic emission spectroscopy (iCAP7400, Thermo Fisher Scientific, USA). The foulant solution was also freeze-dried for determining the valence of the Fe element with XPS (Escalab250Xi, Thermo Fisher Scientific, USA).

**Biofouling characterizations in terms of microbial community and metagenomic analysis**
To extract the fouling layer, the membrane sample (~ 5.25 cm²) was cut into pieces and shaken in a 2 mL 0.9% NaCl solution. The relative amount of biomass was determined in terms of ATP content using the BacTiter-Glo™ ATP kit (Promega (Beijing) Biotech Co., Ltd.). The relative values of intracellular ROS production, SOD enzyme activity, and CAT enzyme activity of microbes were detected using the Reactive Oxygen Species Assay Kit (DCFH-DA, Beytime, China), Superoxide Dismutase Activity Assay Kit (WST−1 Method, Solarbio, China), and Catalase Activity Assay Kit (Solarbio, China), respectively.

To reveal the dynamic shift of microbial community structure, the feed water and fouled RO membrane samples were collected for DNA extraction and 16S rDNA high-throughput sequencing. The concentration and purity of DNA were measured by 1% agarose gel electrophoresis and NanoDropOne. Primers 515 F and 806 R were used to amplify the V4 region of the 16S rRNA gene[64]. PCR products were purified using the E.Z.N.A. Gel Extraction Kit (Omega, USA), and a gene library was prepared using the Next Ultra DNA Library Prep Kit (NEB, USA). Sequencing was then performed using the Illumina HiSeq 2500 platform. To further elucidate the effects of coagulant types on the metabolic processes of the microbiome, the metagenomes of the 20 d membrane samples in the Ctrl, Al, and Fe groups were sequenced on the Illumina HiSeq 2500 platform using the 150-bp paired-end strategy (Magigene, Guangdong).

The α- and β-diversity analyses were implemented for exploring the disparities in microbial community among the three groups. The ecological interaction networks were established via the molecular ecological network analysis pipeline (http://ieg4.rccc.ou.edu/mena) and were visualized using Gephi 0.10.1 (WebAtlas, France). The functional annotation and abundance analysis of metabolic pathway were performed using the Kyoto Encyclopedia of Genes and Genomes database (https://www.genome.jp/kegg/).

### Statistical analysis
Multiple linear regression in conjunction with variance partitioning analysis and principal component analysis was utilized to quantify the relative contributions of biofouling, organic fouling, and inorganic fouling[65] (see detailed calculating schematic in Supplementary Fig. 20). The FQ value was calculated by a one-tailed non-parametric Wilcoxon signed rank test, and the regions of above-moderate significance ($p < 0.1$) were plotted on the Ex/Em map to highlight the difference in EEM spectrum between different foulants. Statistical analysis of the remaining data was conducted using one-way analysis of variance (ANOVA) by SPSS Statistics 26 (IBM, USA). The significance was denoted as follows: ns, $p \geq 0.05$; *, $p < 0.05$; **, $p < 0.01$; ***, $p < 0.001$.

### Reporting summary
Further information on research design is available in the Nature Portfolio Reporting Summary linked to this article.

## Data availability
The sequencing data are available in the NCBI database named BioProject PRJNA1059279. Source data supporting the findings of this study are provided with this paper. Source data are provided in this paper.

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

## Acknowledgements

This work was supported by the Major Program of MOST of China (No. 2022YFC3203103 to X.H.), Beijing Natural Science Foundation (No. JQ22027 to K.X.), Fundamental Research Funds for the Central Universities (QNTD202506 to S.L.), Huaneng Group science and technology research project (HNKJ22-H105 to X.H. and S.L.), and China Postdoctoral Science Foundation (2025M781191 to H.D.).

## Author contributions

X.H., S.L., and H.D. conceived the idea and designed the research. H.D. carried out the experiments. R.G. and Y.L. contributed to data collection. H.D., W.L., and C.C. analyzed and integrated the data. H.D. wrote the initial draft. Y.F.L. contributed to experimental design and manuscript review. H.D. and S.L. iterated multiple versions. X.H., S.L., K.X., and H.D. made the final revision of the manuscript.

## Competing interests

The authors declare no competing financial interest.
