## [Transparent Peer Review file · Nature Communications]

Coagulation pretreatment could deteriorate reverse osmosis membrane fouling

Corresponding Author: Professor Xia Huang

Version 0:

Reviewer comments:

Reviewer #4

(Remarks to the Author)

This paper explores how coagulation pretreatment influences reverse osmosis (RO) membrane fouling in zero liquid discharge (ZLD) brine management systems. By comparing iron- and aluminum-based coagulants, the authors systematically investigate how residual metals in the feed water affect fouling layer composition, microbial communities, and overall RO performance. Their work finds detailed chemical and biological mechanisms that explain why coagulation, while intended to reduce fouling, can in fact worsen it when residues are not adequately removed.

Pretreatment for RO in ZLD applications is a timely and highly important topic because stable RO operation is essential for achieving sustainable water reuse and meeting increasingly strict discharge regulations. With industries under pressure to recover more water and minimize liquid waste, understanding how pretreatment impacts RO performance is critical. This study provides valuable new insights into how common coagulation practices can unintentionally intensify fouling, making it a meaningful and useful contribution to the literature on ZLD brine management and advanced water treatment.

The practical relevance of this work to full-scale ZLD operations could be strengthened. In the experiments, the RO feed was filtered using a 0.45 μm membrane, but in real systems, ultrafiltration or tight microfiltration membranes with much smaller pore sizes are more typical for pretreatment. This difference may affect the amount and type of residual coagulant reaching the RO membranes and therefore the severity of fouling. The authors should discuss these limitations and strongly recommend that actual ZLD plants employ sand filters or tight UF membranes as pretreatment to reduce coagulant carryover and better protect RO membranes in real-world applications.

Reviewer #6

(Remarks to the Author)

This study bridges the gap between lab scale and industrial application of RO technology in Zero liquid discharge of industrial wastewater and provides the guidance for RO fouling control by coagulation pretreatment. The authors have addressed all comments satisfactorily and I recommend this paper for publication.

Response to Reviewers' Comments and List of Changes Made in the Manuscript

REVIEWER #4

Comments:

Summary:

This paper explores how coagulation pretreatment influences reverse osmosis (RO) membrane fouling in zero liquid discharge (ZLD) brine management systems. By comparing iron- and aluminum-based coagulants, the authors systematically investigate how residual metals in the feed water affect fouling layer composition, microbial communities, and overall RO performance. Their work finds detailed chemical and biological mechanisms that explain why coagulation, while intended to reduce fouling, can in fact worsen it when residues are not adequately removed.

Pretreatment for RO in ZLD applications is a timely and highly important topic because stable RO operation is essential for achieving sustainable water reuse and meeting increasingly strict discharge regulations. With industries under pressure to recover more water and minimize liquid waste, understanding how pretreatment impacts RO performance is critical. This study provides valuable new insights into how common coagulation practices can unintentionally intensify fouling, making it a meaningful and useful contribution to the literature on ZLD brine management and advanced water treatment.

Our Response:

We sincerely thank the reviewer for the highly positive and insightful comments on our work. We are grateful for the recognition of the novelty and practical relevance of this study, particularly the systematic elucidation of how residual coagulants can unintentionally deteriorate RO membrane fouling in ZLD systems. We also appreciate the reviewer's acknowledgement that this work provides meaningful mechanistic insights into the coupled chemical and biological processes governing RO fouling under coagulation pretreatment, which we believe will be valuable for both fundamental understanding and practical ZLD operation.

Reviewer's Comment 1:

The practical relevance of this work to full-scale ZLD operations could be

strengthened. In the experiments, the RO feed was filtered using a 0.45 µm membrane, but in real systems, ultrafiltration or tight microfiltration membranes with much smaller pore sizes are more typical for pretreatment. This difference may affect the amount and type of residual coagulant reaching the RO membranes and therefore the severity of fouling. The authors should discuss these limitations and strongly recommend that actual ZLD plants employ sand filters or tight UF membranes as pretreatment to reduce coagulant carryover and better protect RO membranes in real-world applications.

Our Response:

We thank the reviewer for this constructive and insightful suggestion. The adoption of 0.45 µm membrane pretreatment was based on field investigations of actual coal-fired power plant ZLD systems (microfiltration units with relatively loose pore sizes in the range of 0.3–0.9 µm are still commonly employed^{1–6}, particularly in existing or retrofitted plants where operational simplicity and cost considerations dominate), with the aim of simulating real-world ZLD treatment scenarios. We fully agree that ultrafiltration and conventional sand filtration should be recommended more and increasingly adopted in ZLD applications, as they can more effectively reduce particulate matter and coagulant residues prior to RO.

Following the reviewer’s suggestion, we have accordingly expanded the discussion on this limitation and its engineering implications in the revised manuscript. We have also emphasized that tighter pretreatment, such as sand filtration or ultrafiltration, should be recommended in practical ZLD applications to minimize coagulant carryover, thereby reducing the risk of residual metal-induced RO fouling. The revised manuscript now reads as:

Lines 438–448: “It should be noted that this study deliberately focused on a low-pressure microfiltration pretreatment scenario, which was selected to represent practical operating conditions commonly encountered in existing industrial ZLD systems, under which residual coagulants are more likely to be carried over to the RO stage and induce fouling challenges. While this configuration captures realistic and representative fouling behaviors, future work is warranted to further evaluate how coagulation pretreatment influences RO membrane fouling in ZLD systems employing tighter pretreatment configurations, such as ultrafiltration or conventional sand filtration. At the same time, the present findings potentially indicate that adopting sand filtration or tight ultrafiltration to minimize coagulant carryover can be a recommended strategy for improving RO stability in practical RO-based ZLD applications.”

References cited in this response:

1. Li, X., Wu, H., Zhao, J. & Yue, X. An example of zero-discharge project design of high-concentration saline wastewater from a coal chemical industrial park (in Chinese). *Ind. Water & Wastewater* **56**, 79–83 (2025).
2. X. Drake, M., Willer, G., Venkatadri, R., Wise, S. & Charan, N. Treatment of cooling tower blowdown water with membranes in a zero liquid discharge (ZLD) power plant. *Proc. Int. Water Conf.* **73**, 12–15 (2012).
3. US Environmental Protection Agency. Water Reuse Action Plan – Appendix H: Selected Case Studies. (2019).
4. Wang, H., Dai, R., Wang, L., Wang, X. & Wang, Z. Membrane fouling behaviors in a full-scale zero liquid discharge system for cold-rolling wastewater brine treatment: a comprehensive analysis on multiple membrane processes. *Water Res.* **226**, 119221 (2022).
5. Liu, M. et al. Research and demonstration on reclamation of chemical industrial wastewater with high salinity and hardness and purification of reverse osmosis concentrates. *Desalination* **551**, 116437 (2023).
6. Sui, Z., Guo, Y., Sun, G., Zhu, Q. & Liang, S. Zero discharge engineering example of the factory integrated wastewater of Tianjin Iron & Steel Group Co., Ltd. (In Chinese). *Ind. Water Treat.* **33**, 85–88 (2013).

REVIEWER #6

Comments:

Summary:

This study bridges the gap between lab scale and industrial application of RO technology in Zero liquid discharge of industrial wastewater and provides the guidance for RO fouling control by coagulation pretreatment. The authors have addressed all comments satisfactorily and I recommend this paper for publication.

Our Response:

We sincerely thank the reviewer for the positive evaluation of our manuscript and for recommending it for publication. We appreciate the reviewer's recognition of the relevance of this work in bridging laboratory-scale mechanistic understanding with industrial ZLD practice.